# Expanding Mouse-Adapted Yamagata-like Influenza B Viruses in Eggs Enhances In Vivo Lethality in BALB/c Mice

**DOI:** 10.3390/v14061299

**Published:** 2022-06-14

**Authors:** Matthew J. Pekarek, Erika M. Petro-Turnquist, Adam Rubrum, Richard J. Webby, Eric A. Weaver

**Affiliations:** 1Nebraska Center for Virology, School of Biological Sciences, University of Nebraska-Lincoln, Lincoln, NE 68583, USA; mpekarek2@huskers.unl.edu (M.J.P.); epetro-turnquist2@huskers.unl.edu (E.M.P.-T.); 2Department of Infectious Diseases, St. Jude Children’s Research Hospital, Memphis, TN 38105, USA; adam.rubrum@stjude.org (A.R.); richard.webby@stjude.org (R.J.W.)

**Keywords:** influenza B virus, mouse-adapting, growth system, embryonated eggs, hemagglutinin, neuraminidase

## Abstract

Despite the yearly global impact of influenza B viruses (IBVs), limited host range has been a hurdle to developing a readily accessible small animal disease model for vaccine studies. Mouse-adapting IBV can produce highly pathogenic viruses through serial lung passaging in mice. Previous studies have highlighted amino acid changes throughout the viral genome correlating with increased pathogenicity, but no consensus mutations have been determined. We aimed to show that growth system can play a role in mouse-adapted IBV lethality. Two Yamagata-lineage IBVs were serially passaged 10 times in mouse lungs before expansion in embryonated eggs or Madin–Darby canine kidney cells (London line) for use in challenge studies. We observed that virus grown in embryonated eggs was significantly more lethal in mice than the same virus grown in cell culture. Ten additional serial lung passages of one strain again showed virus grown in eggs was more lethal than virus grown in cells. Additionally, no mutations in the surface glycoprotein amino acid sequences correlated to differences in lethality. Our results suggest growth system can influence lethality of mouse-adapted IBVs after serial lung passaging. Further research can highlight improved mechanisms for developing animal disease models for IBV vaccine research.

## 1. Introduction

Influenza viruses are respiratory pathogens that cause up to 5 million severe cases of illness annually [1]. Influenza B viruses (IBVs), whose primary reservoir is humans [2,3], are one of two types of influenza viruses that cause annual epidemics of varying severity [1]. IBVs are responsible for roughly 25% of all influenza cases on an annual basis, but seasonal epidemics can approach or even surpass influenza A viruses (IAVs) in total and severe infections [4,5,6]. IBVs are categorized into two antigenically and genetically distinct lineages, B/Victoria- (B/Vic) and B/Yamagata-like (B/Yam) based on the surface glycoprotein, hemagglutinin (HA). These lineages have cocirculated since the late 1970s or early 1980s [7] and both cause infections each influenza season today.

IBVs have been shown to lead to significant disease just as severe as IAVs [8,9] and often cause heightened disease burden in children [10,11]. In 2013–2014, a quadrivalent influenza vaccine was recommended for yearly influenza vaccines in the US for increased protection against IBV and is still recommended in the United States today [12]. Additionally, recent findings suggest that the two lineages are undergoing constant changes in their antigenic epitopes, and B/Vic viruses experienced major antigenic changes in the hemagglutinin around 2018 [6,13,14,15]. The results from clinical, epidemiological, and evolutionary research all point to the need for more attention on both molecular research and further development of novel vaccine and therapeutic technologies to protect against IBV.

Historically, influenza viruses have been grown in the allantoic fluid of embryonated eggs as the primary vaccine production strategy [16,17]. However, the discovery of Madin–Darby canine kidney (MDCK) cell susceptibility to influenza viruses [18] has further opened studies into cell-culture-based vaccines [19,20,21]. Today, these two methods of virus production are the most common methods for inactivated or live-attenuated vaccine development. However, IBVs are not consistently found infecting species outside humans [2,3], which then requires adaptation of the viruses to develop a disease model for in vivo research. Scientists have used serial lung passaging, termed mouse-adapting, of virus to study viral characteristics or to increase pathogenicity of IBVs [22,23,24,25,26] and IAVs [27,28,29] in mice. Mouse-adapted strains are valuable for vaccine and therapeutic research by causing severe infection, which can highlight differences in protection after vaccination or therapeutic delivery. However, this process is very time consuming and resource intensive. Some studies reporting the mouse-adaptation of IBVs expanded the virus to high titer in eggs [22,23]. Alternatively, many more recent reports of mouse-adapted IBVs opt to grow the viruses in MDCK cell culture after passage in mice [24,25,26]. Amino acid mutations from all eight viral gene segments have been associated with increased pathogenicity after mouse-adapting [23,24,25,26], but none of the observed mutations were shared across these studies. Further understanding of the characteristics leading to more lethal IBV infections in mice will provide better tools for use in IBV vaccine and therapeutic research.

In this report, we set out to determine if the growth system of mouse-adapted IBVs after serial lung passaging influences lethality in mice. We used two Yamagata-lineage viruses, B/Phuket/3073/2013 and B/Florida/4/2006, as our parent viruses due to their inclusion in previously licensed influenza vaccines. Interestingly, we found that after 10 serial lung-lung passages of both parent strains, virus expanded in eggs was more lethal in BALB/c mice than the same virus expanded in MDCK (London line) (MDCK-Ln) cell culture. This effect was also seen after additional passaging of B/Florida/4/2006, even though the additional passaging did not further increase the lethality of the virus. We further identified that the effect on lethality was not due to differences in surface glycoprotein amino acid sequence of B/Florida/4/2006 due to no observed changes in the hemagglutinin (HA) or neuraminidase (NA) protein sequence. Overall, these results show that growing mouse-adapted B/Yam strains in eggs may produce virus with higher lethality in mice than expansion in cell culture.

## 2. Materials and Methods

### 2.1. Cells and Virus Stocks

The following reagents used in this study were kindly provided by the International Reagent Resource (IRR): influenza B virus B/Phuket/3073/2013 (Phu/13) [FR-1364] and the Madin–Darby canine kidney cells, London line (MDCK-Ln) [FR-58]. The following virus was kindly provided by the Biodefense and Emerging Infectious Disease Repository (BEI): influenza B virus B/Florida/4/2006 (FL/4/06) [NR-9696]. Stock viruses obtained from IRR and BEI were used to infect the allantoic cavity 10 or 11 day-old embryonated chicken eggs (Charles River Laboratories, Wilmington, MA, USA) and incubated at 33–35 °C. Allantoic fluid was harvested 48–72 h after infection and virus was quantified through hemagglutinating units (HAU), aliquoted, and stored at −80 °C. Each virus was only subjected to one round of growth in embryonated eggs after arrival from IRR or BEI. MDCK-Ln cells were maintained in Dulbecco’s Modified Essential Medium (DMEM) (Cytiva) supplemented with 5% fetal bovine serum (FBS) and 1% penicillin/streptomycin (P/S) (Cytiva) antibiotics and incubated at 37 °C in 5% CO_2_.

### 2.2. Mice

The 6–8 week old female BALB/cJ mice were purchased from the Jackson Laboratory (Bar Harbor, ME, USA). Mice were housed for experiments and general care at the University of Nebraska-Lincoln (UNL) Life Science Annex on campus according to the Association for Assessment and Accreditation of Laboratory Animal Care (AAALAC) guidelines. The protocols followed for the experiments described were approved by the UNL Institutional Animal Care and Use Committee (IACUC) project number 2158. All experiments were conducted according to the guidelines provided in the Public Health Service Animal Welfare Policy, Animal Welfare Act, the NIH Guide for the Care and Use of Laboratory Animals, and policies put forth by the UNL animal care facility staff and veterinarians.

### 2.3. Hemagglutination Assay

A total of 50 µL sterile Dulbecco’s phosphate buffered saline (DPBS) (HyClone, Logan, UT, USA) was added to rows 2–12 of a 96-well V-bottom plate (Gibco). An amount of 100 µL of virus sample was added to row 1 in duplicate and serially diluted two-fold down the plate. In total, 50 µL of 0.5% chicken red blood cells (cRBCs) (Lampire Biologicals, Piperville, PA, USA) were added to each well. Virus was incubated at room temperature with cRBCs for 30–45 min before reading agglutination patterns [30] in each well to determine HAU of each virus.

### 2.4. Influenza B Virus Mouse-Adaptation

Female BALB/cJ mice (n = 2) were anesthetized i.p. with ketamine/xylazine (K/X) (92.5/7.5 mg/kg K/X) prior to infection. Mice were then infected with 20 µL thawed allantoic fluid from one of the two parent viruses. Three days post-infection, mice were weighed and sacrificed for harvest of the lungs. Lungs were manually homogenized in DPBS and centrifuged to separate the homogenized lysate from the cellular debris. An amount of 30 µL lung lysate from each mouse was combined for a uniform sample. In total, 20 µL of this combined lung lysate was used to infect the next group (n = 2) of mice to begin passage 2. This process was repeated an additional nine times to complete 10 immediate lung-lung passages (p10). For B/FL/4/06, the process was repeated using allantoic fluid from eggs infected with p10 lung lysate as the stock virus and proceeded to complete a total of 20 lung-lung passages (p20). A model summarizing the process is shown (Figure 1).

### 2.5. Mouse-Adapted Virus Growth

Lung lysate from each virus was used to infect both 10–11-day old embryonated chicken eggs and a 25 cm^2^ tissue-culture flask (Falcon) of MDCK-Ln cells. Infected eggs were incubated at 33 °C for 72 h before being placed at 4 °C overnight. The next day, allantoic fluid from each egg was harvested and HAU were quantified before storing at −80 °C. MDCK-Ln cells in a 25 cm^2^ flask with 2 µg/mL (0.002%) TPCK-treated trypsin in the media were infected and incubated at 33 °C with 5% CO_2_ for 48 h. Cell supernatant was then collected and centrifuged in order to remove cell debris. The supernatant was used to then infect a 75 cm^2^ tissue culture flask seeded with MDCK-Ln cells and incubated for 48 h 33 °C with 5% CO_2_. After 48 h, the supernatant was again collected and centrifuged before infecting a 225 cm^2^ tissue culture flask of MDCK-Ln cells for 72 h before a final supernatant harvest. All strains grown in MDCK-Ln cells were verified to have an HAU ≥ 256 before harvest to ensure efficient virus production and cytopathic effects (CPE) were observed during incubation at each sequential step. Virus from both eggs and cells were quantified using HAU, aliquoted, and stored at −80 °C until future use.

### 2.6. Tissue Culture Infectious Dose (TCID_50_)

A total of 10 µL of virus was diluted 1:10 in the first row of a 96-well U-bottom plate. The initial dilution was then serially diluted 10-fold down the plate. An amount of 1.5 × 10^5^ MDCK-Ln cells in DMEM supplemented with 5% FBS without antibiotics were then added into each well and the cell-virus mixture was incubated at 33 °C with 5% CO_2_ for 24 h. After incubation, the plate was washed twice using DPBS to remove free virus from the wells and 200 µL DMEM without FBS or antibiotics with 2 µg/mL TPCK-treated trypsin was added to all the wells. The plates were then incubated for 72 h at 33 °C with 5% CO_2_ before adding 50 µL of 0.5% cRBCs into each well. The agglutination patterns were analyzed 45 min after addition of cRBCs [30]. TCID_50_ values were calculated using the Reed–Muench method [31] and obtained from three independent experiments.

### 2.7. Reverse Transcription-Polymerase Chain Reaction (RT-PCR)

RNA was extracted from allantoic fluid or cell-free supernatant containing virus using the PureLink Viral RNA/DNA Mini Kit (Invitrogen, Waltham, MA, USA) by following the manufacturer’s recommendations. The viral RNA was then run through qPCR using the Luna Universal Probe One-Step RT-qPCR Kit (New England Biolabs, Ipswich, MA, USA) using the Universal Influenza B Primer Probe Set (BEI Resources, Manassas, VA, USA) [NR-15608, NR-15609, NR-15610]. RT-PCR was performed using the QuantStudio 3 Real-Time PCR System (Applied Biosystems) with the following parameters: 55 °C for 30 min, 95 °C for 2 min, 95 °C for 15 s, and 60 °C for 15 s as recommended by BEI resources for the influenza virus real-time RT-PCR assay. A total of 40 cycles were used to complete amplification. IBV positive samples were identified by a Ct value difference of ≥10 cycles above the no template control reactions.

### 2.8. Influenza Challenges

Upon anesthetization with K/X, female BALB/cJ mice (n = 5) were infected with specified logTCID_50_ unit viral challenge in a 20 µL dose. Mice were then followed for 14 days post-infection to compare weight loss between groups or humanely sacrificed upon loss of 25% initial weight. For parent virus infection (p0), allantoic fluid containing either Phu/13 or FL/4/06 was diluted 1:2 in DPBS for challenge with a 1:100 dilution of the TCID_50_/mL previously calculated. For mouse-adapted viruses, dilutions were calculated based off delivering 10-fold serial dilutions from 6log_10_TCID_50_ to 3log_10_TCID_50_ units in 20 µL doses.

### 2.9. RNA Isolation, Amplification and Sequencing

IBV RNA was generated from 200 µL of virus stock using the Qiagen RNeasy mini kit (Cat: 74104). The RNA was eluted in RNA-free water and converted to cDNA by one-step reverse transcription and PCR using a Superscipt III RT kit (Invitrogen; Cat: 12574035). Primers (10 µM) B-HANA-UniF (5′-GGGGGGAGCAG AAGCAGAGC-3′) and B-HANA-UniR (5′-CGGGTTATTAGTAGTAACAAGAGC-3′), 25 µL reaction master mix, 16µL nuclease-free water, 5 µL of RNA, and 2 µL SSIII RT/Plat Taq was combined. cDNA was amplified using the ProFlex PCR system by Applied Biosystems. The HA and NA amplified PCR product was gel extracted using the Qiagen Gel extraction kit (Cat: 28706) and eluted in 40 µL elution buffer. An amount of 2 µL of cDNA was combined with 1 µL B specific oligos (3.2 µM) and 9 µL water and submitted to the St. Jude Children’s Research Hartwell Center for Sanger sequencing. Ab1 files received by the Hartwell center were combined and analyzed with DNASTAR Lasergene 15.

### 2.10. Statistical Analysis

GraphPad Prism v9 was used to complete all data analysis. Infectivity data shown represent averages of three biological replicates with the standard deviation between the trials. In vivo challenge study data is the average of 5 mice included in each group. Error bars represent the standard deviation at each data point. *p* values below 0.05 were considered significant in this study. Statistical analysis of TCID_50_ titers was performed through multiple unpaired t-tests using two-tailed *p* values, and significant differences in survival were inferred using the log-rank Mantel–Cox to determine differences between groups (* *p* < 0.05, ** *p* < 0.01).

## 3. Results

### 3.1. Lethality of Phu/13 and FL/4/06 in Mice Prior to Mouse-Adapting

Influenza viruses do not typically cause disease in mice. However, lethal challenge strains of influenza are often optimal for vaccine and therapeutics studies [26,29,32,33,34]. Therefore, mouse-adapting is used to produce a disease model for use in laboratory studies [23,26]. Before beginning the mouse-adapting process, we first needed to characterize any pathogenicity these viruses already possessed in mice. TCID_50_/mL virus titers were calculated before challenging female BALB/cJ mice with a 1:2 dilution of chorioallantoic fluid containing either 3.5log_10_TCID_50_ units Phu/13 or 5log_10_TCID_50_ units FL/4/06 in a 20 µL dose (Figure 2A). This was the maximum infectious dose of each virus that could be delivered. Weight loss was monitored for 14 days after infection (Figure 2B). No signs of meaningful weight loss were detected in the two weeks post-challenge with the maximum amount of virus able to be used prior to mouse-adaptation.

### 3.2. Infectivity of maB/Yam Viruses in Vitro and Lethality in Mice after 10 Passages

To increase lethality for use in challenge studies, serial lung-passaging of these viruses was performed. A detailed schematic is shown (Figure 1). Direct lung-lung passages were performed without an intermediate growth step in between to maintain the virus in the ultimate target setting of the mouse lung. After completion of 10 serial lung-lung passages, the homogenized lung lysate was used to infect both embryonated eggs and MDCK-Ln cell culture to produce large volumes of high-titer viruses for future use. RT-PCR was then used to confirm presence of IBV RNA after expansion (data not shown). The resultant virus stocks were denoted using the strain combined with the passage number and growth medium (p10E or p10C). Before infecting mice with the newly expanded mouse-adapted viruses, we determined in vitro infectivity of the viruses using MDCK-Ln cells (Figure 3A). Both viruses grew to high infectious titers after expansion of virus from the lung lysate. Interestingly, FL/4/06 p10C was less infectious in vitro than p10E despite being grown in the cells used for the infectivity assay (Figure 3A).

Groups of five mice were challenged with p10E or p10C from either strain using the same TCID_50_ unit dose to determine if either expansion media conferred higher mortality in vivo. Each group was infected with a serial 10-fold dilution starting from 6log_10_TCID_50_ units and were monitored for weight loss and mortality. A table of the median lethal dose (MLD_50_) values calculated with the Reed–Muench method [31] is provided (Figure 3B). The full weight loss graph for each MLD_50_ is provided (Figure 3C,D). Groups are presented based on whether the virus was grown in eggs (p10E) or in cells (p10C) and the log_10_ TCID_50_ challenge dose. Weight loss was much more drastic in mice infected with p10E strains down to as low as a 5log_10_ TCID_50_ unit dose when compared with virus grown in cells. Similar weight loss was observed between p10E and p10C of both FL/06 and Phu/13 at a 4log_10_ TCID_50_ unit dose, below our calculated MLD_50_ values. When we focused on the 5log_10_ TCID_50_ unit dose, we observed a significant difference in lethality of both FL/06 and Phu/13 p10E when compared to p10C (Figure 3C,D). Together, these results show after 10 serial passages of two different strains of B/Yam, amplification of virally infected lung lysate in eggs produced a significantly more lethal virus in mice compared to the same lung lysate amplified in cell culture.

### 3.3. Infectivity in Vitro and Lethality in Mice of FL/4/06 after 10 Additional Passages

We also set out to see what role additional passaging may play on one of our strain’s virulence and if it supports our observations from 10 direct lung-lung passages. To do this, the FL/4/06 p10E virus was used to complete another round of 10 lung-lung passages as described above. The egg-passaged virus was chosen due to its higher initial pathogenicity in mice. After an additional 10 passages, the homogenized lung lysate was used to infect embryonated eggs and MDCK-Ln cell culture for viral expansion. TCID_50_/mL values were once again confirmed and used for dosing in the challenges. Again, FL/4/06 p20E grew to a higher infectious titer in MDCK-Ln cell culture than p20C (Figure 4A).

The challenge doses used for p20E and p20C were identical infectious titers to p10E and p10C. MLD_50_ calculations were repeated using the p20 strains (Figure 4B). Again, the post-infection weight loss and survival curves (Figure 4C) show that the p20E viruses maintained lethality significantly better after expansion than p20C viruses. Significant weight loss was observed by day 8 in mice infected with the highest doses of p20E while no severe weight loss was observed in mice infected with any dose of p20C. In the 6 log_10_ TCID_50_ dose, infection with p20E led to significantly more mortality than infection with p20C (Figure 4C). These results corroborate what was seen after 10 passages of both FL/06 and Phu/13. Interestingly, neither p20E nor p20C viruses increased in lethality after 10 additional passages as measured through MLD_50_ titers (Figure 3B and Figure 4B). Differences in survival between p10E and p20E were not significant (Figure 4D). This is especially notable because FL/06 p10E was used to start the further 10 passages, and no significant weight loss or lethality was observed after the virus was grown in MDCK-Ln cells. This highlights the potential for further study to understand why this cell culture system is not conducive to the growth of mouse lethal IBV strains.

### 3.4. Sequence Analysis of FL/4/06 HA and NA before and after Mouse-Adapting

To attempt to find any genetic mutations responsible for the lethality differences observed between the parental and mouse-adapted viruses, we sequenced the HA and NA genes of FL/4/06. We chose to sequence this virus since it had been serially passaged 20 total times in mice. We observed that the HA and NA of the p10 and p20 FL/4/06 were identical to the parental virus irrespective of growth system. The four major antigenic sites of the influenza B virus HA are shown (Figure 5A) [35]. The only identifiable difference observed after sequencing was a single nucleotide mutation in the HA sequence (Figure 5B). However, this mutation was silent and did not change the amino acid sequence. The complete HA amino acid and nucleotide sequence alignments are shown in Appendix A, respectively. A representative NA amino acid sequence alignment from positions 161–320 is shown (Figure 5C). All amino acid sequences were identical to the parental virus prior to mouse-adapting. The full-length NA amino acid and nucleotide alignments are shown in Appendix A, respectively.

## 4. Discussion

Influenza B viruses are relevant respiratory pathogens that cause significant disease burden each year around the globe [1,4,5,6]. Despite this, the inability to naturally cause infections in many mouse strains has led to difficulty of developing easily accessible small animal disease models for IBV vaccine and therapeutic research. The process of mouse-adapting IBV has been used to define in vivo pathogenic determinants [24,25,26] and for development of lethal strains for use in vaccine and therapeutics studies [32,33,34,36]. However, performing mouse-adapting series can be costly and may not produce consistent or repeatable results. Differing genetic determinants of pathogenicity studies in the BALB/c mouse model [23,24,25,26] also can contribute to a lack of consistency between procedures. Thus, development of a widely available reverse genetics system for generating lethal viruses such as those for IAV [37,38] is less feasible. Recognizing these issues with the mouse-adapting process, we designed this study to investigate whether growth setting can influence the lethality of IBV after serial lung-lung passaging in mice (Figure 1).

Our results suggest that viral growth system may play a role in mouse-adapted IBV lethality. Having been initially grown in eggs, the parent virus strains showed no lethality in mice over two weeks of infection (Figure 2). These viruses were then passaged 10 times in mice as described above and amplified in both embryonated eggs and MDCK-Ln cells for further study. A total of 10 initial passages were chosen based on previous studies in which mouse-adapted strains of IBV [23] as well as various IAV subtypes were deemed sufficient to increase lethality of the viruses [39,40]. This was also performed more recently for swine influenza H1 viruses [41]. Since 10 passages is often fewer than is required to mouse-adapt IBV [24,25,26], this served as our benchmark for potential improvement on established methods from the literature. Our model also eliminated any intermediate growth steps in embryonated eggs or MDCK-Ln cell culture in between lung passages. We believed that this may improve pathogenicity at the end after viral growth by maximizing the amount of time spent in mouse lung tissue and preventing any mutations driven by replication in other systems.

As a proxy for overall pathogenicity, we determined the median lethal dose for the p10 mouse-adapted B/Yam strains grown in eggs and cells. Mice were challenged with equivalent TCID_50_ units of each virus (Figure 3). These mice were followed 14 days for weight loss and survival (Figure 3C,D), and the MLD_50_ was calculated (Figure 3B). We observed that expanding the final passage lung lysate in eggs led to increased weight loss and significantly lower chance of survival than expansion in MDCK-Ln cell culture for both B/Yam strains. While not often considered in research studies, preparation of mouse-adapted B/Yam stocks appears to play a role in the lethality of these viruses in mice.

To further support our findings, we took one of the two lethal mouse-adapted B/Yam strains and completed a further round of 10 serial lung-lung passages. FL/4/06 p10E was chosen as the initial strain for infection due to its higher initial lethality seen in mice. After 20 total lung-lung passages, the final lung lysate was once again used to infect embryonated eggs or MDCK-Ln cell culture. Infectious titers were obtained in vitro (Figure 4A) and used to calculate challenge doses in mice. Once again, at the same infectious dose, FL/4/06 p20E led to increased weight loss and decreased chance of survival when compared to p20C (Figure 4C). Interestingly, neither p20E nor p20C were able to increase the level of lethality in mice as measured by the MLD_50_ titer compared to p10E, the strain used to start further mouse-adapting. While the difference in lethality of p10E and p20E was found to be insignificant (Figure 4D), p20C appeared to lose almost all lethality even after 10 additional passages. This was an unexpected result, but it is possible that the mutations needed to further increase the lethality did not occur in the 10 additional passages. It is also possible that the intermediate egg growth interfered with the ability of these mutations to occur. Additionally, we saw no amino acid mutations in either of the surface glycoproteins even after 20 total serial lung passages. Previous studies have identified mutations across many of the different gene segments, but internal gene mutations are often shown to have the most critical effect on viral pathogenicity [23,24,26]. Further sequencing of the entire viral genome will help elucidate exactly what mutations occurred in our model.

MDCK-Ln cells were chosen for this study due to the increased growth rate and reported increased susceptibility to influenza virus infection with the ability to produce virus at a higher titer than MDCK cells [42,43]. While these strains have both previously been incorporated as influenza vaccine strains, two representative strains may not represent all the diversity in the Yamagata-lineage. More work is required to determine if this phenomenon is shared not only among other strains of B/Yam, but also of B/Vic and ancestral IBV. Another consideration to make is the presence of defective-interfering particles. These particles have been shown to play active roles in modulating host immune responses while also inhibiting replication of normal infectious particles [44]. Our serial lung-lung passage model is unable to rule out production of these particles or highlight what role they could play in the process if these particles are present.

It is also important to recognize egg growth of IBV may not be appropriate for certain studies. Both IBV and IAV have been shown to undergo mutations in the HA which shift receptor-binding capability towards embryonated eggs and away from human epithelial tissue by passaging [45,46]. These egg-adapted mutations also have been shown to alter the antigenicity of the viruses [47]. However, our viral glycoprotein sequence analysis showed that even over 20 lung passages as well as expansion steps in embryonated eggs and/or cell culture, no further mutations occurred in either the HA or NA. This would suggest the single expansion step did not introduce any egg-adaptation mutations after serial lung passaging. This could be impactful for some vaccine development studies to show limited development of egg-adapted mutations using single replication steps rather than repeated egg passage over time. Additionally, if growth of virus in eggs was responsible for lethality, we would have expected to see some mortality in our parent strain challenge at a high concentration of egg-grown virus (Figure 2B). This leads us to conclude that the serial lung-lung passaging played the primary role in the lethality we observed.

In this study, our preliminary findings support that influenza B virus pathogenicity in mice may not require adaptation in the surface glycoproteins to increase lethality. While our results cannot rule out internal gene mutations, any genetic differences between virus grown in eggs and virus grown in cell culture would have been influenced by the growth systems. These different settings appear to be capable of influencing the in vivo lethality of mouse-adapted IBV. Further research with our model will better characterize this phenomenon and assist in overcoming a major hurdle in the influenza vaccine and therapeutic research field. By increasing the availability of better disease models, we hope to assist in the development of more suitable tools to study this important disease.

## 5. Conclusions

Influenza B viruses are primarily a human pathogen. While known to cause a significant portion of annual influenza infections and severe disease, development of mouse disease models has historically been difficult and inconsistent. We aimed to identify if the growth system of viral expansion after mouse-adapting influences lethality when used in mouse challenge studies. We used two Yamagata-lineage strains for our mouse-adapting model. Our results show that at the same in vitro infectious dose, virus grown in embryonated eggs is significantly more lethal than the same virus grown in MDCK-Ln cell culture. The differences in lethality between virus grown in eggs and cell culture was not due to amino acid changes in the surface glycoprotein sequences. Further studies will aim to identify the mechanism which drives these differences in lethality.

## Figures and Tables

**Figure 1 viruses-14-01299-f001:**
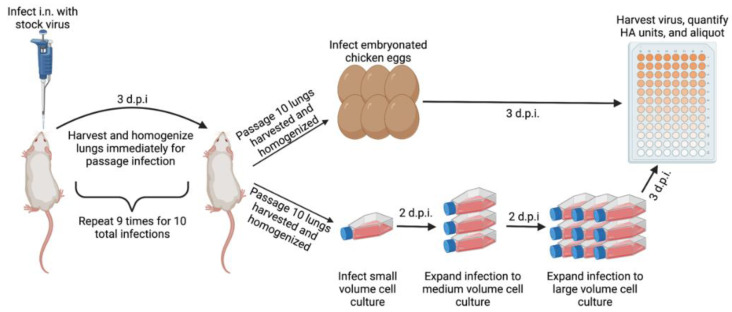
Schematic depicting mouse-passaging procedure. Passage 1 mice were infected i.n. with 20 µL of stock virus. In 3 d.p.i., the lungs were harvested and homogenized before pooling of the lung lysate. Passage 2 mice were then infected with 20 µL of pooled lung lysate and the procedure was repeated up to passage 10. After homogenizing passage 10 lungs, the lysate was diluted and used to infect either embryonated chicken eggs or MDCK-London cells. Infected embryonated eggs were incubated at 33 °C for 72 h before being moved to 4 °C overnight. Allantoic fluid was harvested the next day. An amount of 25 cm^2^ infected MDCK-London cells were incubated at 33 °C for 48 h before clarified supernatant was used to infect a 75 cm^2^ flask. The 75 cm^2^ flask was incubated for 48 h at 33 °C before expansion into a 225 cm^2^ flask. Finally, 225 cm^2^ MDCK-London cells were incubated at 33 °C for 72 h before harvesting virus from supernatant. Schematic created in BioRender.

**Figure 2 viruses-14-01299-f002:**
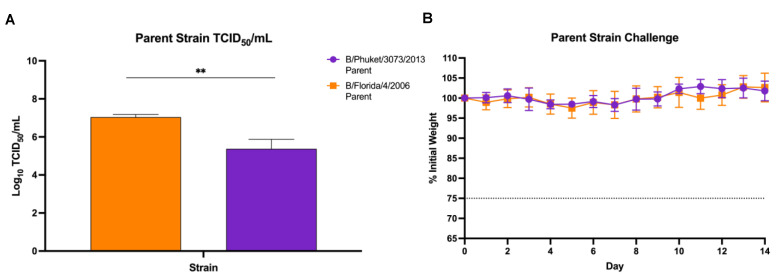
Live challenge with wild-type B/Florida/4/2006 and B/Phuket/3073/2013. (**A**) TCID_50_ values for each strain used for initial passage infection. Values shown are an average of three independent experiments with error bars representing standard deviation between three independent replicates. Statistical significance was determined through unpaired *t*-test, ** *p* < 0.01; (**B**) Weight loss over 14 days shown after i.n. infection with 1:2 dilution of allantoic fluid of each parent virus. A ≥25% initial weight lost (dotted line) was used as a threshold for humane sacrifice of the animals.

**Figure 3 viruses-14-01299-f003:**
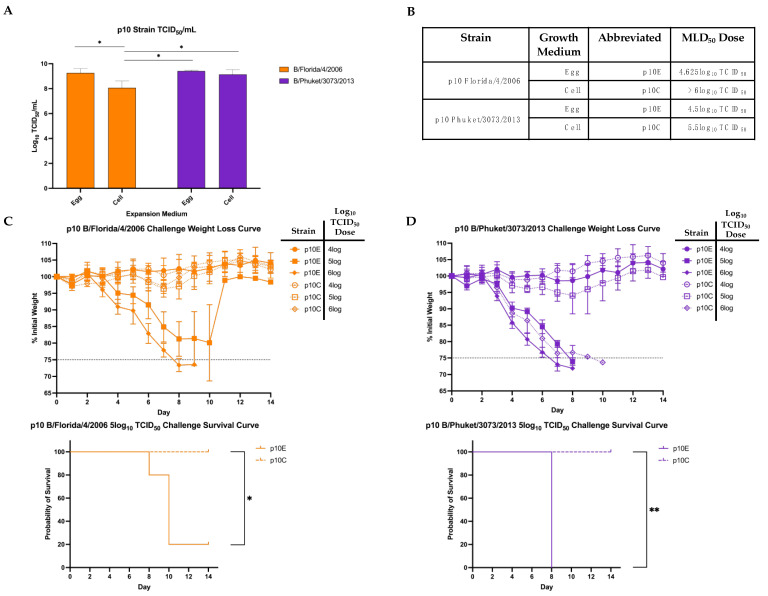
Determination of pathogenicity of maB/Yam in mice. (**A**) TCID_50_/mL titers of both strains after 10 lung-lung passages and expanded in either embryonated eggs (p10E) or MDCK-Ln cell culture (p10C). (**B**) Table breaking down MLD_50_ values calculated using the Reed–Meunch method (31) for each virus strain. (**C**) Mice challenged with doses corresponding to logTCID_50_ of FL/4/06 p10E or p10C were followed for 14 d.p.i. for weight loss (top) and survival after challenge with 5log TCID_50_ units (bottom). A ≥25% initial weight lost was used as a threshold for humane sacrifice of the animals. (**D**) Mice challenged with doses corresponding to logTCID_50_ of Phu/13 p10E or p10C were followed for 14 d.p.i. for weight loss (top) and survival after challenge with 5log TCID_50_ units (bottom). A ≥25% initial weight lost was used as a threshold for humane sacrifice of the animals. Statistical differences in TCID_50_/mL titers were calculated using unpaired t-tests, while statistical significance between group survival was calculated through log-rank Mantel–Cox test. Statistical analysis was performed in GraphPad Prism 9.0 (* *p* < 0.05, ** *p* < 0.01).

**Figure 4 viruses-14-01299-f004:**
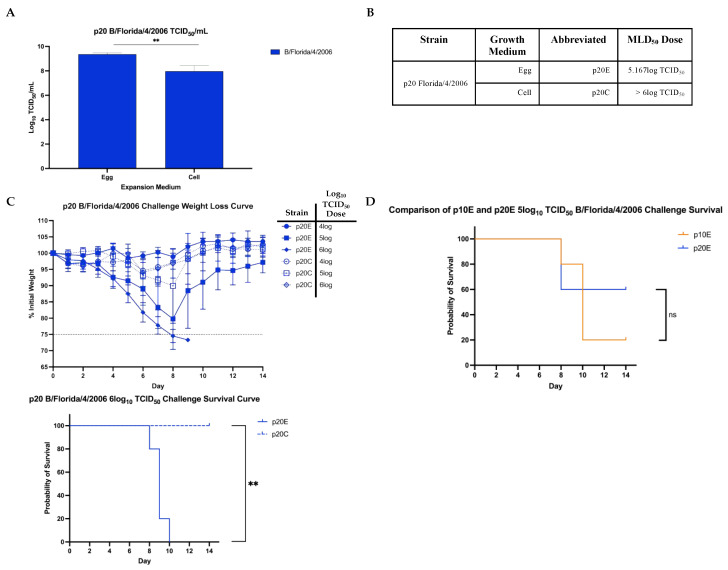
Pathogenicity of FL/4/06 after 20 total lung-lung passages. (**A**) TCID_50_/mL titers of FL/4/06 p20E and p20C. (**B**) Table breaking down MLD_50_ values calculated using the Reed–Meunch method (31) for FL/4/06 p20E and p20C. (**C**) Mice challenged with doses corresponding to logTCID_50_ of FL/4/06 p20E or p20C were followed for 14 d.p.i. for weight loss (top) and survival after challenge with 6log TCID_50_ units (bottom). A ≥25% initial weight lost was used as a threshold for humane sacrifice of the animals. (**D**) Comparison of survival probability of mice challenged with 5log TCID_50_ of p10E or p20E. No statistically significant difference in survival probability was observed. Statistical differences in TCID_50_/mL titers were calculated using unpaired t-tests, while statistical significance between group survival was calculated through log-rank Mantel–Cox test. Statistical analysis was performed in GraphPad Prism 9.0 ** *p* < 0.01).

**Figure 5 viruses-14-01299-f005:**
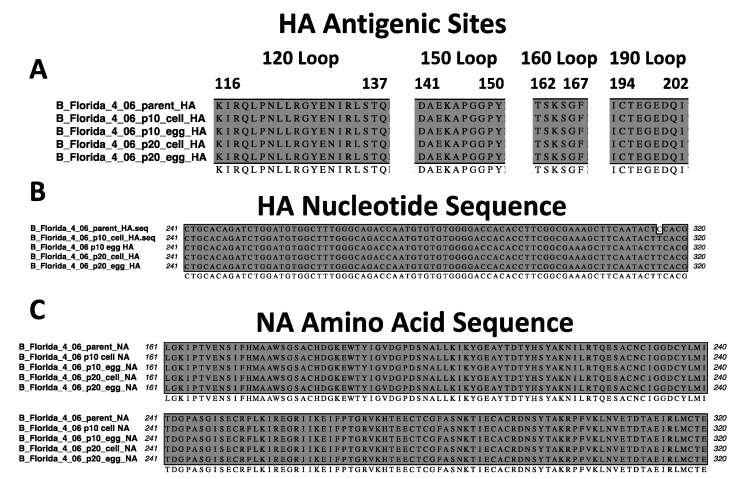
FL/4/06 surface glycoprotein sequence analysis. The HA and NA genes were reverse transcribed into cDNA, PCR amplified, and sequenced using Sanger sequencing. (**A**) The major antigenic sites [35] of the HA protein are shown to be identical to the parental virus. (**B**) The single nucleotide mutation observed between the parental virus and the mouse-adapted HA sequences was a single silent mutation between nucleotide 241 and 320. (**C**) A representative amino acid alignment of the NA protein from positions 161–320 shows 100% sequence conservation between the parental and the mouse-adapted viruses.

## Data Availability

No new data were created or analyzed in this study. Data sharing is not applicable to this article.

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
