# Peer review of "Expanding Mouse-Adapted Yamagata-like Influenza B Viruses in Eggs Enhances In Vivo Lethality in BALB/c Mice"

_viruses, 2022, doi:10.3390/v14061299_

Round 1
Reviewer 1 Report
The authors have now sequenced their mice massaged viruses, which adds knowledge to their study. The manuscript is improved and I feel it should now be accepted for publication.
Reviewer 2 Report
Dear Editor,
I carefully read through the authors response and revised manuscript and found it is nice. Authors took into account all comments and significantly improved the article. I don't feel qualified to judge about the English language and style. However the study can be accepted after Editorial office reviewing
Sincerely,
Just a comment: I am not clear enough about part when U-shaped plates used. Authors originally had it written as if the culture was conducted in plates with a U-shaped bottom, but in fact they conducted experiments using cell culture flask.. U-shaped plates likely were used for HI test
Reviewer 3 Report
The proposed work is devoted to an interesting topic - obtaining a strain of human influenza virus type B adapted to mice. The results obtained by the authors are of interest to virologists, microbiologists, and especially for specialists involved in the development and study of the effectiveness of vaccines. The article is written in clear and clear language, all conclusions are confirmed by the results of experiments. I believe that the article can be accepted for publication without changes.
This manuscript is a resubmission of an earlier submission. The following is a list of the peer review reports and author responses from that submission.
Round 1
Reviewer 1 Report
Influenza B viruses of both the Victoria- and Yamagata-lineages are implicated in a large proportion of the morbidity and mortality associated with influenza outbreaks. Influenza B viruses have been exclusively considered as human pathogens, although they were shown to infect seals. The animal models available for studying influenza B viruses are limited (Bouvier and Lowen, 2010; Thangavel and Bouvier, 2014). Influenza B virus infection in ferrets is generally mild (Kim et al., Viral Immunol, 2009; Huang et al. PLoSOne, 2011) but certain strains may cause severe clinical disease (Huang et al., J. Gen. Virol , 2014). Influenza B viruses of both the Victoria- and Yamagata-lineages were studied in BALB/c and pharmacologically immunosuppressed BALB/c mice (Marathe et al., Antiv Res, 2017) and in immunocompromised BALB scid and NOD scid mice (Pascua et al., Sci Rep, 2017). These studies showed that B/Brisbane/60/2008 virus is capable to replicate in mouse lungs and cause weight changes in BALB/c mice without prior adaptation.
Adaptation of influenza viruses to a mouse model is a well-established approach to obtain variants with enhanced virulence in mammalian host. Overall, these studies showed that multiple molecular determinants (sequence changes) within HA and the ribonucleoprotein (RNP) complex of influenza viruses contribute to their adaptation and increased virulence in mice. In the current study, the authors conducted 10 serial lung-to-lung passages in mice, grow the stocks in two host systems (embryonated chicken eggs and MDCK-London cells) and determined pathogenicity in mice of these stocks.
Comments:
- The major conclusion of the authors that growth environment can play a role in mouse-adapted influenza B virus pathogenicity required additional experiments and further investigation. The authors did not provide any data on why culturing influenza B viruses in MDCK-London cells affected pathogenicity. Further characterization of receptor specificity of generated mouse-adapted variants is required to understand binding abilities to α2,3 and α2,6 sialic acid-linked receptors. It is likely that the changes in receptor specificity contributed not only to the increased initial ability of the virus to infect and spread in the lung bronchiolar epithelium but also to the broader tissue tropism in other major mouse organs. Therefore, ability of mouse-adapted variants to replicate in other organs should be explored.
- Determination of molecular markers of enhanced virulence of mouse-adapted influenza B viruses must be done. Therefore, additional experiments are required to determine amino acid substitutions in mouse-adapted variants and whether these substitutions are stable after culturing in eggs and MDCK-London cells. Additionally, egg-adapted HA substitutions may be responsible for enhanced virus pathogenicity in a mouse model. This must be shown.
- The pathogenicity of initial virus stocks and mouse-adapted variants must be conducted with a range of virus doses/inputs. This is not always done in the current study and therefore it is difficult to compare the data obtained with the P0 and P10 viruses of Victoria- and Yamagata-lineages.
- Representation of the data is unclear and require major changes. The authors should always present the data for initial virus stocks, P10 variants, P20 variants, P10-egg, P10-MDCK.
- Experimental approach contains many short-cuts. For example, the authors used undiluted mouse lung suspension after each passage to inoculate to the next passage. It is possible to suggest that defective viral particles are generated, and this must be proven or disproven.
- Lack of novelty of the current study is a major concern.
Author Response
Reviewer 1
Influenza B viruses of both the Victoria- and Yamagata-lineages are implicated in a large proportion of the morbidity and mortality associated with influenza outbreaks. Influenza B viruses have been exclusively considered as human pathogens, although they were shown to infect seals. The animal models available for studying influenza B viruses are limited (Bouvier and Lowen, 2010; Thangavel and Bouvier, 2014). Influenza B virus infection in ferrets is generally mild (Kim et al., Viral Immunol, 2009; Huang et al. PLoSOne, 2011) but certain strains may cause severe clinical disease (Huang et al., J. Gen. Virol , 2014). Influenza B viruses of both the Victoria- and Yamagata-lineages were studied in BALB/c and pharmacologically immunosuppressed BALB/c mice (Marathe et al., Antiv Res, 2017) and in immunocompromised BALB scid and NOD scid mice (Pascua et al., Sci Rep, 2017). These studies showed that B/Brisbane/60/2008 virus is capable to replicate in mouse lungs and cause weight changes in BALB/c mice without prior adaptation.
Adaptation of influenza viruses to a mouse model is a well-established approach to obtain variants with enhanced virulence in mammalian host. Overall, these studies showed that multiple molecular determinants (sequence changes) within HA and the ribonucleoprotein (RNP) complex of influenza viruses contribute to their adaptation and increased virulence in mice. In the current study, the authors conducted 10 serial lung-to-lung passages in mice, grow the stocks in two host systems (embryonated chicken eggs and MDCK-London cells) and determined pathogenicity in mice of these stocks.
Comments
- The major conclusion of the authors that growth environment can play a role in mouse-adapted influenza B virus pathogenicity required additional experiments and further investigation. The authors did not provide any data on why culturing influenza B viruses in MDCK-London cells affected pathogenicity. Further characterization of receptor specificity of generated mouse-adapted variants is required to understand binding abilities to α2,3 and α2,6 sialic acid-linked receptors. It is likely that the changes in receptor specificity contributed not only to the increased initial ability of the virus to infect and spread in the lung bronchiolar epithelium but also to the broader tissue tropism in other major mouse organs. Therefore, ability of mouse-adapted variants to replicate in other organs should be explored.
Changing the receptor binding specificity is possible during the adapting process. However, if egg-adapted mutations were responsible for this, we feel the initial stocks would have shown some pathogenicity. Discussion was added to address this as follows: “Additionally, if growth of virus in eggs was solely responsible for pathogenicity, we would have expected to see some morbidity in our parent strain challenge at a high concentration of egg-grown virus (Figure 2B). This leads us to conclude that the serial lung-lung passaging has played a role in the pathogenicity observed in our results.” While we are unable to identify a specific mechanism of increased pathogenicity with our current study, we have shown that virus which is expanded in embryonated eggs leads to increased pathogenicity when used to infect mice through lethality tests. In the future, we hope to identify one or more mechanisms which are responsible for these results. Currently, this is outside the scope of our initial study.
- Determination of molecular markers of enhanced virulence of mouse-adapted influenza B viruses must be done. Therefore, additional experiments are required to determine amino acid substitutions in mouse-adapted variants and whether these substitutions are stable after culturing in eggs and MDCK-London cells. Additionally, egg-adapted HA substitutions may be responsible for enhanced virus pathogenicity in a mouse model. This must be shown.
We are very interested in determining the exact mechanism of our observations. At this time, we have been unable to perform the whole genome RNA sequencing which would be required to address this concern. We discuss some egg-adapted mutations in the HA in the discussion regarding the vaccine development field. This is a future direction we are planning to pursue but is outside the scope of our current study.
- The pathogenicity of initial virus stocks and mouse-adapted variants must be conducted with a range of virus doses/inputs. This is not always done in the current study and therefore it is difficult to compare the data obtained with the P0 and P10 viruses of Victoria- and Yamagata-lineages.
Only one dilution was used for the challenge with initial stocks due to low pathogenicity that has previously been described using IBV in the BALB/c mouse model. We thought the undiluted would be the best chance to see any morbidity with a challenge. It is very rare that viruses are more pathogenic at lower doses. However, this is a possibility and will be considered in future studies.
- Representation of the data is unclear and require major changes. The authors should always present the data for initial virus stocks, P10 variants, P20 variants, P10-egg, P10-MDCK.
Figures 2-4 were updated to try to clarify their organization. Additional text was also added in the results section to attempt to better focus which strains are being highlighted.
- Experimental approach contains many short-cuts. For example, the authors used undiluted mouse lung suspension after each passage to inoculate to the next passage. It is possible to suggest that defective viral particles are generated, and this must be proven or disproven.
Our logic for performing direct lung-lung passages was that the less time the virus spends in an in vitro environment, the better chance we have of ending up with pathogenic virus at the end of the process. This was addressed in the results section with “Direct lung-lung passages were performed without an intermediate growth step in between to maintain the virus in the ultimate target setting of the mouse lung.” We also expanded on this in the discussion section with “Our model also eliminated any intermediate growth steps in embryonated eggs or MDCK-Ln cell culture in between lung passages. We hypothesized that this may improve pathogenicity at the end after viral growth by maximizing the amount of time spent in mouse lung tissue and preventing any mutations driven by replication in other environments.”
The defective particle topic was an interesting concern that we had not previously considered. Unfortunately, the turnaround requested for revisions means we are unable to properly address this experimentally. We added a section with our other limitations in the discussion which reads “Another consideration to make is the presence of defective-interfering particles. These particles have been shown to play active roles in modulating host immune responses while also inhibiting replication of normal infectious particles [43]. Our serial lung-lung passage model is unable to rule out production of these particles or highlight what role they could play in the process if these particles are present.”
- Lack of novelty of the current study is a major concern.
While mouse-adapting is a well-established procedure in the literature, we are unaware of a published study which directly compares growth environment after mouse-adaptation and the effect on pathogenicity. This was highlighted in the introduction section with the following text: “To our knowledge, no previous study has directly compared the effect of viral growth in embryonated egg and Madin-Darby canine kidney (MDCK) cell lines using mouse-adapted IBV.”
We also believe that our results can help the vaccine and therapeutic development field improve their success at developing lethal IBV disease models in mice. This is highlighted in the revised discussion with “Our results could improve the ease and consistency of developing lethal strains for use in challenge studies, which are often used as a strong measure of protection for many pre-clinical vaccine and therapeutic trials [32-35].”
Reviewer 2 Report
The presented work is of great interest to virologists studying the influenza virus. The authors were able to adapt the type B influenza virus to mice and achieve a lethal infection. Obtaining such an adapted version of the influenza B virus makes it possible to study the possibilities of infection with this virus of other animal species other than humans. The article is written in clear language, well illustrated, and I believe that it can be accepted for publication without modification.
Author Response
Reviewer 2
The presented work is of great interest to virologists studying the influenza virus. The authors were able to adapt the type B influenza virus to mice and achieve a lethal infection. Obtaining such an adapted version of the influenza B virus makes it possible to study the possibilities of infection with this virus of other animal species other than humans. The article is written in clear language, well-illustrated, and I believe that it can be accepted for publication without modification.
We thank you for your supportive comments. We appreciate your time for reviewing and your support of our manuscript.
Reviewer 3 Report
The manuscript entitled “Expanding Mouse-Adapted Yamagata-like Influenza B Viruses in Eggs Maintains in vivo Pathogenicity” by Pekarek and colleagues aims at assessing IBV pathogenicity determinants from mouse adapted viruses passaged in eggs or cells. While the study is of interest, two big caveats are to me of concern and I have the following suggestions and questions:
Main points:
- While viruses passaged 10 times in mice are re-passaged once in eggs, they are re-passaged 3 times in MDCK cells before their mouse pathogenicity is re-evaluated. The cell adaptation is thus more likely than the egg adaptation. The authors should compare identical passages in the 2 “environments” to be consistent.
- The viruses were not sequenced nor after 10 (or 20) passages in mice, nor after eggs/cells passages, which is a real pity. Mutations associated with mouse/egg/cell adaptation are likely to be seen and they would explain the results observed here. Sequencing (of HA for the least, full genomes ideally) would really allow for comparing with previous studies and for drawing hypotheses on determinants of adaptation as well as of pathogenicity for these IBV.
Minor points:
- Lines 46-48, I think the authors make a link that should not be made here. Ferrets and hamsters seem good animal models for the study of IBV as well.
- Why did the authors sometimes culture eggs/cells at 37°C, other times at 33-35°C, other times at 33°C? This is confusing.
- Eggs passages: did the authors infect the chorioallantoic membranes? Or the allantoic fluid?
- Which TPCK concentration was used? (0,0002% is unclear as we would need to know the concentration of the stock solution)
- Lines 145-147 would need rephrasing; they are unclear as such.
- Figure 2: where were 2 different doses used for the 2 IBV?
- After 20 passages in mice the Florida virus seems to have been attenuated (the MLD50 is lower). This is interesting and should be discussed.
Author Response
Reviewer 3
The manuscript entitled “Expanding Mouse-Adapted Yamagata-like Influenza B Viruses in Eggs Maintains in vivo Pathogenicity” by Pekarek and colleagues aims at assessing IBV pathogenicity determinants from mouse adapted viruses passaged in eggs or cells. While the study is of interest, two big caveats are to me of concern and I have the following suggestions and questions:
Main points:
While viruses passaged 10 times in mice are re-passaged once in eggs, they are re-passaged 3 times in MDCK cells before their mouse pathogenicity is re-evaluated. The cell adaptation is thus more likely than the egg adaptation. The authors should compare identical passages in the 2 “environments” to be consistent.
The reason virus was passaged 3 times in cell culture is to obtain enough volume to use as a challenge virus for study, and if only 1 passage in cells is attempted, we are unable to rescue enough virus to use for large challenge studies. Future investigation aims to passage each stock in the other medium to serve as a “cross expansion” and see what effect this has on pathogenicity, if any. This may help address this concern, but we feel this is outside the scope of our original study.
The viruses were not sequenced nor after 10 (or 20) passages in mice, nor after eggs/cells passages, which is a real pity. Mutations associated with mouse/egg/cell adaptation are likely to be seen and they would explain the results observed here. Sequencing (of HA for the least, full genomes ideally) would really allow for comparing with previous studies and for drawing hypotheses on determinants of adaptation as well as of pathogenicity for these IBV.
We are very interested in determining the exact mechanism of our observations. At this time, we have been unable to perform the whole genome RNA sequencing which would be required to address this concern. We discuss some egg-adapted mutations in the HA in the discussion regarding the vaccine development field. Once we are able to get this data, we aim to pursue a reverse genetics system to see if introduction of these mutations into other B/Yam viruses leads to an increase in pathogenicity without having to perform serial lung passaging.
Minor points:
- Lines 46-48, I think the authors make a link that should not be made here. Ferrets and hamsters seem good animal models for the study of IBV as well.
This statement was removed.
- Why did the authors sometimes culture eggs/cells at 37°C, other times at 33-35°C, other times at 33°C? This is confusing.
Uninfected cells were maintained at 37ËšC. Growth of initial stock viruses was done at 33Ëš or 35ËšC depending on the strain. All mouse-adapted strains were grown at 33ËšC. The different temperatures were used for optimal virus growth/cell maintenance.
- Eggs passages: did the authors infect the chorioallantoic membranes? Or the allantoic fluid?
The allantoic fluid was used as the infection site of the egg and was harvested to create the virus stocks.
- Which TPCK concentration was used? (0,0002% is unclear as we would need to know the concentration of the stock solution)
The final concentration of TPCK-trypsin was 2µg/mL and was revised in the methods to say this.
- Lines 145-147 would need rephrasing; they are unclear as such.
This was revised to state the following: “IBV positive samples were identified by a Ct value difference of ≥ 10 cycles above the no template control reactions.”
- Figure 2: where were 2 different doses used for the 2 IBV?
Two different doses were used for the parent challenge to attempt and characterize if the stocks induced any pathogenicity prior to passaging. The 1:2 dilution was done to normalize the TCID50 units in the 20 µL dose, but otherwise no dilutions were made to the samples. We thought this would maximize the chance to see any pathogenicity prior to serial lung passaging.
- After 20 passages in mice the Florida virus seems to have been attenuated (the MLD50 is lower). This is interesting and should be discussed.
Discussion was added which now reads: “This was an unexpected result, but it is possible that the mutations needed to further increase the mouse pathogenicity did not occur in the 10 additional passages. It is also possible that the intermediate egg growth interfered with the ability of these mutations to occur. This result does help support the idea that additional research into the IBV mouse-adapting process could help increase the efficiency and success of the process.”
Reviewer 4 Report
To Editor, Authors
No doubt, the manuscript Expanding Mouse-Adapted Yamagata-like Influenza B Viruses in Eggs Maintains in vivo Pathogenicity” is of interest for virologists. And also for molecular epidemiologists of influenza viruses and vaccine production authorities. Authors analyzed model for Influenza B virus.
The main goal is to find model animals for the study of vaccines and antivirals. In recent years, the Vic. Lineage has undergone antigenic changes in the NA so the vaccine needs to be updated. And for this we need model animals. Vaccines have always been developed on the Embryonated chicken eggs. Then MDCK was developed for the production of flu vaccines. However, influenza B is almost always found only in humans, so it was not easy to find a model organism. For these purposes, the researchers passaged the virus into the lungs of mice. But this is a long and time-consuming process. 55 passages on mice, then on MDCK cells. After mice, all pathogenic strains acquired mutations, but no mutations were found common to all strains. In this regard, the authors hypothesize that the environment in which viruses grow can determine/influence the nature/localization of the mutation resulting from adaptation to a new host.
Actually data obtained needs to be considered when deciding on prevention and control. Introduction and aims are clear and logic.
But still has some questions to be addressed.
General Comments:
- Chapter 2.4. How did you choose which passage to stop at when passing through the lungs of mice (why exactly 10)? Did you evaluate clinical manifestations, the presence of a virus in the lung tissue during the passage on mice?
- Chapter 2.5. Has the presence of Cytopathic Effect (CPE) in cell culture been evaluated? Or conducted a hemaggutination test? How was the virus detected? Perhaps such conditions for the cultivation of the virus you used were not optimal, the productivity of the cultivation could be insufficient.
- Chapter 2.6. Why were U-shaped tablets used to titrate the virus in cell culture, and not flat-bottomed tablets used for cell culture? In this case, the distribution of cells on the surface of the well will be uneven.
- Why wasn't the CPE evaluated? In some cases 72 hours is not enough for the appearance of CPE.
- I didn't see how the authors explain the main result (their assumption about why strains are more pathogenic according to the mouse–chicken embryo-mouse scheme). The authors in the text say that the cultivating on chicken embryo can shift the ability of the virus to bind from human cells to chicken cells, citing links, but this is about the limitations of the method, and not about the intended explanation of the result.
- I didn't quite understand the logic – please follow these parts: “… The process of mouse-adapting IBV has been used to define in vivo pathogenic determinants [24- 26] and is often performed to develop lethal strains… However, it is costly and not assured of working before starting the procedure… To overcome these obstacles we performed direct lung-lung passaging in BALB/c mice to produce mouse-adapted IBV strains…”
- Why didn't you provide sequencing of the original strain, 10p, 20p and 10p, cultivated in chicken embryos and cells? It is possible that the using one or another cultivation system has led to substitutions in the genetic determinants that determine the pathogenicity of the virus.
Specific comments:
- Line 79: you use “Chorioallantoic fluid” without reference. If you address to the reference “30” – please check if it is allantoic fluid. What is correct one?
- Lines 141-143: It seems to be a good idea to add a methodology / publication where the use of primers is described.
- Line 202: "RT-PCR was then used to confirm the presence of IBV RNA after expansion (data not shown)" - but the authors in the methods highlighted the item "2.7. Quantitative Polymerase Chain Reaction (qPCR)". Was there a difference in the amount of virus produced on cells and on embryos? Just why do you describe the item "Quantitative PCR" if PCR was used only for “…to confirm presence..."?
- 184 – 1:2 dilution was used for infection, however in line 195 - 1:100 dilution was used for infection (for the same experiment)
- Lines 258-261 neither p20E nor p20C vi- 258 ruses were able to reach the same level of pathogenicity in mice, as measured through MLD50 titers, as the initial p10E used for additional passaging (Figure 3B, 4B). Differences in survival between p10E and p20E were not significant (Figure 4D). It seems to me that the phrases contradict each other.
Author Response
Reviewer 4
No doubt, the manuscript Expanding Mouse-Adapted Yamagata-like Influenza B Viruses in Eggs Maintains in vivo Pathogenicity” is of interest for virologists. And also for molecular epidemiologists of influenza viruses and vaccine production authorities. Authors analyzed model for Influenza B virus.
The main goal is to find model animals for the study of vaccines and antivirals. In recent years, the Vic. Lineage has undergone antigenic changes in the NA so the vaccine needs to be updated. And for this we need model animals. Vaccines have always been developed on the Embryonated chicken eggs. Then MDCK was developed for the production of flu vaccines. However, influenza B is almost always found only in humans, so it was not easy to find a model organism. For these purposes, the researchers passaged the virus into the lungs of mice. But this is a long and time-consuming process. 55 passages on mice, then on MDCK cells. After mice, all pathogenic strains acquired mutations, but no mutations were found common to all strains. In this regard, the authors hypothesize that the environment in which viruses grow can determine/influence the nature/localization of the mutation resulting from adaptation to a new host.
Actually data obtained needs to be considered when deciding on prevention and control. Introduction and aims are clear and logic.
General Comments:
- Chapter 2.4. How did you choose which passage to stop at when passing through the lungs of mice (why exactly 10)? Did you evaluate clinical manifestations, the presence of a virus in the lung tissue during the passage on mice?
This passage number was chosen based on previous reports in the literature trying to match or decrease the number of passages required to mouse-adapt IBV strains and move towards less adaptation which is generally required of IAV. Discussion was added and reads: “We chose to start with 10 initial passages based on previous studies which mouse-adapted strains of IBV [23] as well as various IAV subtypes as their total passage number [38-39]. This was also performed more recently for swine influenza H1 viruses [40]. Since 10 passages is often fewer than is required to mouse-adapt IBV [24-26], this served as our benchmark for potential improvement on established methods from the literature.”
- Chapter 2.5. Has the presence of Cytopathic Effect (CPE) in cell culture been evaluated? Or conducted a hemaggutination test? How was the virus detected? Perhaps such conditions for the cultivation of the virus you used were not optimal, the productivity of the cultivation could be insufficient.
Virus was measured and quantified during virus growth using HAU. With our cutoff measurement of an HAU of ≥ 256, we deemed 72 hours a sufficient growth time for viruses in cell culture. A line was added to the methods section which reads: “All strains grown in MDCK-Ln cells were verified to have an HAU ≥ 256 before harvest in order to ensure efficient virus production.”
- Chapter 2.6. Why were U-shaped tablets used to titrate the virus in cell culture, and not flat-bottomed tablets used for cell culture? In this case, the distribution of cells on the surface of the well will be uneven.
We use U-bottom plates for our infectivity assays to maximize the consistency of our results we read through RBC pelleting. We have seen fairly consistent results across individual trials. All of the virus growth which would be more sensitive to cell distribution is done in canted neck tissue culture flasks. This is better described in section 2.5 to try and eliminate any confusion and a sentence was added to section 2.6 which reads: “By using U-bottom plates, we are able to more clearly interpret RBC pelleting for the infection assay to eliminate discrepancies in reading different experiments.”
- Why wasn't the CPE evaluated? In some cases 72 hours is not enough for the appearance of CPE.
Described above in point 2. Verification of efficient virus growth in cell culture was done through hemagglutination assay and further experiments controlled for infectious doses to ensure equivalent infection takes place.
- I didn't see how the authors explain the main result (their assumption about why strains are more pathogenic according to the mouse–chicken embryo-mouse scheme). The authors in the text say that the cultivating on chicken embryo can shift the ability of the virus to bind from human cells to chicken cells, citing links, but this is about the limitations of the method, and not about the intended explanation of the result.
Discussion was added to state our hypothesis which reads: “By characterizing median lethal doses as a measure of pathogenicity, we have concluded that growth of mouse-adapted IBV in embryonated eggs results in increased pathogenicity at the same infectious dose compared to virus grown in cell culture… While our results are unable to identify a specific mechanism which causes this difference, previous findings in the literature point to a possible genetic influence. We hypothesize that the different growth media after passaging is complete could have induced as little as a single mutation which could be responsible for these differences. Previous studies into IBV pathogenicity in mice have identified mutations in both the viral glycoproteins and internal proteins which were responsible for increased pathogenicity [22-26].”
- I didn't quite understand the logic – please follow these parts: “… The process of mouse-adapting IBV has been used to define in vivo pathogenic determinants [24- 26] and is often performed to develop lethal strains… However, it is costly and not assured of working before starting the procedure… To overcome these obstacles we performed direct lung-lung passaging in BALB/c mice to produce mouse-adapted IBV strains…”
This section set out to describe why the study was performed. The revised section now reads: “The process of mouse-adapting IBV has been used to define in vivo pathogenic determinants [24-26] and for development of lethal strains for use in vaccine and therapeutics studies [32-35]. However, performing mouse-adapting series can be costly and may not produce consistent repeatable results. Differing genetic determinants of pathogenicity studies in mouse models [23-26] also can contribute to a lack of consistency between procedures. Thus, development of a widely available reverse genetics system for generating lethal viruses such as those for IAV [36-37] is less feasible. Recognizing these issues with the mourse-adapting process, we designed this study to investigate whether growth media can influence the pathogenicity of IBV after serial lung-lung passaging in mice (Figure 1).”
- Why didn't you provide sequencing of the original strain, 10p, 20p and 10p, cultivated in chicken embryos and cells? It is possible that the using one or another cultivation system has led to substitutions in the genetic determinants that determine the pathogenicity of the virus.
While we recognize the benefits of including the viral genome sequences in this study. This set of experiments is planned for a future study to determine the exact mechanism which is driving the differences observed in this preliminary study through collaboration with another group or company to perform the sequencing. Discussion was revised to now state: “Further investigation into the whole virus genome sequences will highlight if there is a genetic determinant which has caused these changes in our model, and if so, where the difference(s) occurred. Even if mutations in protein sequences are discovered, it is most likely that these changes were induced during the expansion process in either medium. This would further support our conclusion that growth media can influence the pathogenicity of these strains after mouse-adapting is complete. If no genetic differences are observed, alternative hypotheses need to be considered to determine the causes of increased IBV pathogenicity in mice.”
Specific comments:
- Line 79: you use “Chorioallantoic fluid” without reference. If you address to the reference “30” – please check if it is allantoic fluid. What is correct one?
This was updated in the text to “allantoic fluid.”
- Lines 141-143: It seems to be a good idea to add a methodology / publication where the use of primers is described.
The parameters described were obtained directly from BEI Resources for cycling protocols. The section was revised to now read “RT-PCR was performed using the QuantStudio 3 Real-Time PCR System (Applied Biosystems) with the following parameters: 55ËšC for 30 minutes, 95 ËšC for 2 minutes, 95ËšC for 15 seconds, 60ËšC for 15 seconds as recommended by BEI resources for the influenza virus real-time RT-PCR assay. 40 cycles were used to complete amplification.”
- Line 202: "RT-PCR was then used to confirm the presence of IBV RNA after expansion (data not shown)" - but the authors in the methods highlighted the item "2.7. Quantitative Polymerase Chain Reaction (qPCR)". Was there a difference in the amount of virus produced on cells and on embryos? Just why do you describe the item "Quantitative PCR" if PCR was used only for “…to confirm presence..."?
The section title was updated to now read “2.7 Reverse Transcription-Polymerase Chain Reaction (RT-PCR)” to eliminate any confusion. For this study, we did not complete quantitation of viral genome copy number as a measure of viral infectivity.
- 184 – 1:2 dilution was used for infection, however in line 195 - 1:100 dilution was used for infection (for the same experiment)
This has been fixed in the figure legend to correctly represent the 1:2 dilution which was used for challenge.
- Lines 258-261 neither p20E nor p20C vi- 258 ruses were able to reach the same level of pathogenicity in mice, as measured through MLD50 titers, as the initial p10E used for additional passaging (Figure 3B, 4B). Differences in survival between p10E and p20E were not significant (Figure 4D). It seems to me that the phrases contradict each other.
The lines have been revised and now read “Interestingly, neither p20E or p20C viruses displayed any increase in pathogenicity after 10 additional passages as measured through MLD50 titers (Figure 3B, 4B). Differences in survival between p10E and p20E were not significant (Figure 4D).”
Round 2
Reviewer 1 Report
Lines 17-19 – Major conclusion of the manuscript is unclear and should be changed.
Lines 67-68 – Influenza B/Florida/04/2006 was isolated 16 years ago and cannot be considered as recently isolated. It is also not a component of recently used influenza vaccines.
Lines 78-82 – Provide passage history of IBVs used in the study.
Line 351 – The goal of the study should be carefully stated. It is incorrect to say that the study was designed to investigate whether growth media can influence the pathogenicity of IBV after serial lung-lung passaging in mice. The authors did not study growth medium, they study whether growth of IBVs in different host systems can affect pathogenicity in mice. This should be corrected through out the text of the manuscript (for example, lines 404 and 414).
Reviewer 3 Report
The authors took comments and suggestions into account, resulting in an improved manuscript. The 2 main caveats are still there.